# Reasoning Beyond Points: A Visual Introspective Approach for Few-Shot 3D Segmentation

**Changshuo Wang**
Department of Computer Science
University College London
London, United Kingdom
wangchangshuo1@gmail.com

**Shuting He**
School of Computing and Artificial Intelligence
Shanghai University of Finance and Economics
Shanghai, China
heshuting555@gmail.com

**Xiang Fang**[*]
Interdisciplinary Graduate Programme
Nanyang Technological University, Singapore
xfang9508@gmail.com

**Zhijian Hu**
LAAS
CNRS
Toulouse, France
huzhijian1991@gmail.com

**Jia-Hong Huang**
Information Institute
University of Amsterdam
Amsterdam, Netherlands
j.huang@uva.nl

**Yixian Shen**
Information Institute
University of Amsterdam
Amsterdam, Netherlands
y.shen@uva.nl

**Prayag Tiwari**
School of Information Technology
Halmstad University
Halmstad, Sweden
prayag.tiwari@ieee.org

## Abstract

Point Cloud Few-Shot Semantic Segmentation (PC-FSS) aims to segment unknown categories in query samples using only a small number of annotated support samples. However, scene complexity and insufficient representation of local geometric structures pose significant challenges to PC-FSS. To address these issues, we propose a novel pre-training-free **V**isual **I**ntrospective **P**rototype **Seg**mentation network (**VIP-Seg**). Specifically, we design a Visual Introspective Prototype (VIP) module that employs a multi-step reasoning approach to tackle intra-class diversity and domain gaps between support and query sets. The VIP module consists of a Prototype Enhancement Module (PEM) and a Prototype Difference Module (PDM), which work alternately to progressively refine prototypes. The PEM enhances prototype discriminability and reduces intra-class diversity, while the PDM learns common representations from the differences between query and support features, effectively eliminating semantic inconsistencies caused by domain gaps. To further reduce intra-class diversity and enhance point discriminative ability, we propose a Dynamic Power Convolution (DyPowerConv) that leverages learnable power functions to effectively capture local geometric structures and detailed features of point clouds. Extensive experiments on S3DIS and ScanNet demonstrate that our proposed VIP-Seg significantly outperforms current state-of-the-art methods, proving its effectiveness in PC-FSS tasks. Our code will be available at https://github.com/changshuowang/VIP-Seg .

---

[*]Corresponding author.

39th Conference on Neural Information Processing Systems (NeurIPS 2025).

# 1 Introduction

In recent years, point cloud data has become increasingly important in numerous applications such as autonomous driving [53, 4], robotics [27, 8, 7], and augmented reality [6, 23]. As a fundamental task in 3D scene understanding[11, 10, 32], point cloud semantic segmentation [14, 33, 51, 49] plays a crucial role in these domains. However, acquiring large-scale, high-quality annotated point cloud data demands substantial time and human resources, severely limiting the practical application of traditional deep learning methods.

To address the data scarcity challenge, researchers[16, 43] have turned to few-shot learning for point cloud segmentation tasks . Point Cloud Few-Shot Semantic Segmentation (PC-FSS) [31, 29] aims to segment novel categories in query samples using only a handful of annotated support samples, significantly reducing annotation costs. Zhao et al. [54] pioneered this approach by introducing AttMPTI, based on a pre-trained DGCNN [40]. Subsequent works [19, 13, 55] further enhanced feature extraction and prototype generation strategies, improving performance to some extent. However, PC-FSS faces two major challenges that limit its effectiveness: representation inconsistency within semantic categories and cross-domain feature misalignment. The first occurs when identical semantic categories in support and query samples show significant differences in physical characteristics, such as size, orientation, or visual appearance. Prototypes from support samples aid segmentation of similar objects in query samples but can introduce biases due to these variations. The second challenge involves feature distribution mismatches, where query data contains semantic content not present in the support set, and vice versa.

Moreover, most existing methods (as shown in Fig. 1) rely on pre-training paradigms, which not only increase computational costs but also potentially introduce domain shifts when facing unseen categories, particularly in cross-domain applications. Additionally, the irregular and sparse nature of point clouds makes it challenging to effectively capture local geometric structures, a problem that becomes more pronounced in few-shot scenarios where limited data is available for learning robust representations.

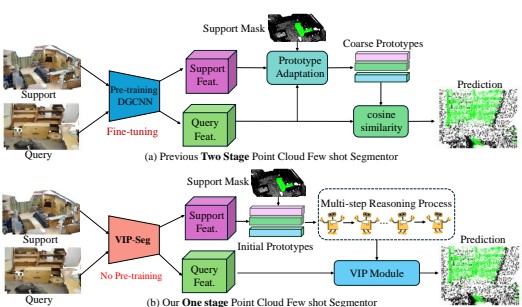

Figure 1: Comparison between previous methods and our approach for PC-FSS. (a) Previous methods typically follow a two-stage pipeline that requires pre-training a DGCNN followed by fine-tuning with prototype adaptation. (b) Our proposed VIP-Seg eliminates the pre-training stage with a single-stage approach that integrates DyPowerConv and employs a multi-step reasoning process to progressively refine prototypes.

To overcome these limitations, we propose VIP-Seg, a novel pre-training-free network for PC-FSS. As shown in Fig. 1, our approach features two key innovations: First, we introduce Dynamic Power Convolution (DyPowerConv), which adaptively models local geometric features by learning region-specific power functions, capturing fine-grained details and structural variations. This enhances the model's ability to distinguish similar structures, further reducing intra-class diversity. Second, we develop a Visual Introspective Prototype (VIP) module to address intra-semantic diversity and domain gaps through a multi-step reasoning approach. The VIP module combines a Prototype Enhancement Module (PEM) and a Prototype Difference Module (PDM) that work alternately to progressively refine prototypes. The PEM improves prototype discriminability through attention mechanisms, while the PDM learns common representations from feature differences, effectively mitigating domain gaps. This iterative process gradually aligns feature distributions and boosts segmentation accuracy.

Our main contributions can be summarized as follows:

- We propose VIP-Seg, a novel pre-training-free framework for point cloud few-shot semantic segmentation that achieves superior performance without time-consuming pre-training.
- We introduce Dynamic Power Convolution, which leverages learnable power functions to adaptively model complex local geometric features, significantly enhancing the network's ability to capture fine-grained structural details.

- We design a Visual Introspective Prototype module that employs a multi-step reasoning approach to effectively address intra-semantic diversity and domain gaps between support and query sets.

- Extensive experiments demonstrate that our approach significantly outperforms state-of-the-art methods across various few-shot settings, proving the effectiveness of our approach.

## 2 Related Works

### 2.1 Point Cloud Semantic Segmentation

Point cloud semantic segmentation [30, 42, 36] is a crucial task in 3D scene understanding [47, 50] that has witnessed significant advancements in recent years. Pioneering works such as PointNet [21] and PointNet++ [22] established the foundation by directly processing point cloud data through multi-layer perceptrons (MLPs). Subsequent research introduced innovative methods leveraging graph convolution, attention mechanisms, and multi-modal approaches. For instance, DGCNN [40] proposed the EdgeConv operation to capture inter-point relationships via dynamically constructed local graphs. Inspired by the extensive progress made by the Transformer architecture in computer vision [25, 24, 37, 39, 34, 18], some researchers are also applying the architecture to point cloud understanding tasks. For example, Point Transformer [52] and its variants [14, 20] incorporated self-attention mechanisms to effectively model long-range dependencies. Recently, several methods based on State Space Models have achieved significant advancements in 3D tasks. Despite these advancements, these methods typically demand substantial annotated data for training, limiting their practical applications.

### 2.2 Point Cloud Few-shot Semantic Segmentation

To tackle the data scarcity challenge in point cloud semantic segmentation, few-shot learning approaches [26] have emerged as a promising solution. Early work by [54] introduced prototype networks to this domain through the AttMPTI method, sparking subsequent research focused on feature enhancement, prototype optimization, and domain adaptation [19, 48, 9, 45, 38, 35]. Recent advancements have developed more sophisticated techniques: [16] incorporated structural information for precise target localization while minimizing background interference, [43] addressed intra-class diversity and semantic inconsistency through bilateral aggregation and consistency purification, and [41] leveraged LLM-generated content to optimize prototypes and mitigate categorical bias. Meanwhile, [2] proposed a novel setting to avoid foreground leakage, while [1] enhanced performance through multi-modal data. Despite these advancements, challenges remain in effectively capturing complex local structures and addressing domain differences.

### 2.3 Dynamic Convolution

Dynamic convolution enhances a model's adaptability and expressive power by generating convolution kernels dynamically based on input data. In 2D image processing, dynamic convolution [46, 15] has been widely adopted for its effectiveness. Inspired by these successes, researchers have extended it to the 3D point cloud domain. [12] proposed DyCo3D, which incorporates dynamic context learning to better capture local point cloud features. [28] developed KPConv, which generates dynamic convolution kernels by learning local geometric structures, while [44] introduced PAConv, leveraging a weight bank and ScoreNet to dynamically assemble convolution kernels, thereby adapting to the irregular structure of point clouds. However, these methods primarily focus on combining multiple convolution kernels through attention coefficients, leaving significant room for improvement in fine-grained semantic understanding of local geometric structures.

## 3 Method

In this section, we first introduce the problem definition of Point Cloud Few-Shot Semantic Segmentation (PC-FSS). Then, we describe the overall architecture of VIP-Seg, as illustrated in Fig. 2. Next, we present the proposed Dynamic Power Convolution (DyPowerConv). Finally, we introduce our Visual Introspective Prototype (VIP) module that employs a multi-step reasoning approach.

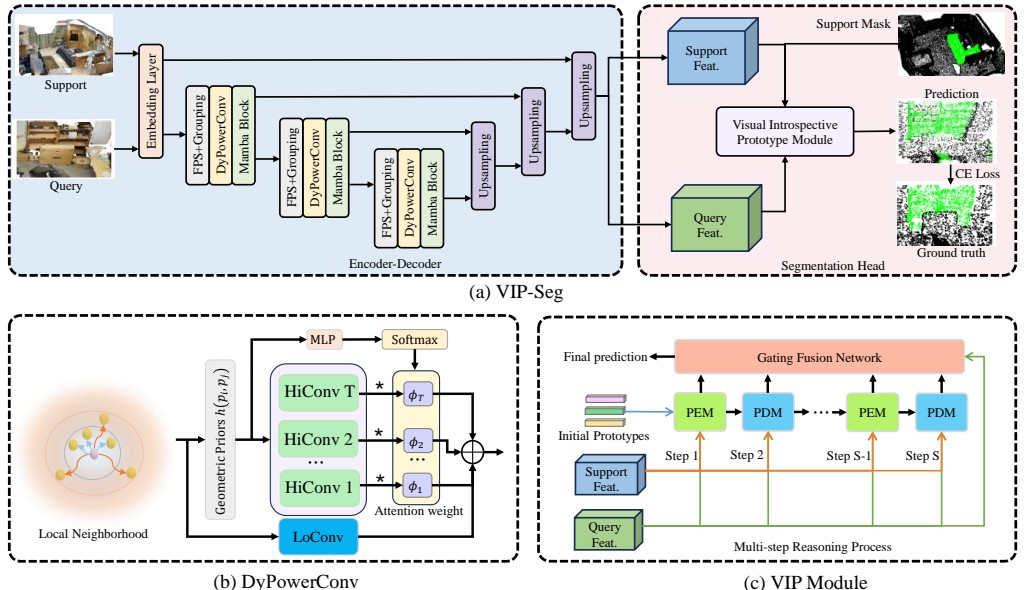

Figure 2: Overview of the proposed VIP-Seg. (a) The overall framework consists of an encoder-decoder backbone for feature extraction and a segmentation head for prototype-based classification. (b) Illustration of our Dynamic Power Convolution (DyPowerConv) module, which combines multiple high-order convolutions (HiConv) with dynamic attention weights to adaptively model local geometric structures. Each HiConv learns a different power function to capture fine-grained details. (c) The Visual Introspective Prototype (VIP) module implements a multi-step reasoning process where the Prototype Enhancement Module (PEM) and Prototype Difference Module (PDM) work alternately to progressively refine prototypes. A gating fusion network integrates predictions from multiple reasoning steps to generate the final segmentation result.

## 3.1 Problem Definition

Following the standard episodic learning paradigm [54], we partition the dataset into two non-overlapping class sets: base classes $\mathcal{C}_{base}$ used for training and novel classes $\mathcal{C}_{novel}$ used for testing, where $\mathcal{C}_{base} \cap \mathcal{C}_{novel} = \emptyset$. The framework operates through N-way K-shot tasks, utilizing paired support and query sets. In each episode, the support set $\mathcal{S} = \{(P_s^{n,k}, M_s^{n,k})\}$ contains $K$ labeled samples for each of the $N$ categories, where $P_s^{n,k} \in \mathbb{R}^{L \times (3+d)}$ represents a point cloud with $L$ points (each having 3D coordinates and d-dimensional features, such as color or surface normals), and $M_s^{n,k} \in \{0, 1\}^L$ denotes the corresponding binary segmentation mask indicating foreground (target class) and background points. The query set $\mathcal{Q} = \{(P_q, M_q)\}$ contains point clouds $P_q \in \mathbb{R}^{L \times (3+d)}$ to be segmented, with ground truth masks $M_q \in \{0, 1, 2, ..., N\}^L$ assigning each point to one of the $N$ target classes or the background class.

## 3.2 Overall Architecture of VIP-Seg

As shown in Fig. 2, we propose VIP-Seg, formulating PC-FSS as a dual optimization problem that combines local structure modeling with progressive prototype refinement. VIP-Seg adopts an encoder-decoder architecture coupled with the proposed VIP module for effective few-shot segmentation. The encoder consists of three stacked DyPowerConv-Mamba blocks that integrate our proposed DyPowerConv for local geometric feature extraction and Mamba blocks [17] for capturing long-range dependencies. DyPowerConv adaptively models local point relationships through learnable power functions, while Mamba blocks efficiently process sequential data with state space models, enabling effective long-range interactions among points. The decoder progressively recovers point cloud resolution through inverse interpolation, propagating features from coarser to finer levels.

### 3.3 Dynamic Power Convolution

To effectively capture local geometric structures and detailed features of point clouds, thereby further reducing intra-class diversity, we propose Dynamic Power Convolution (DyPowerConv). This convolution adaptively models complex local geometric features through learnable power functions, enabling flexible representation of local structures.

#### 3.3.1 Power Function Design

The design of our DyPowerConv is motivated by the need for flexible modeling of local geometric structures in point clouds. We employ learnable power functions to capture features at different scales and levels of detail. DyPowerConv comprises two key components: Low-order Convolution (LoConv) and Dynamic High-order Convolution (DyHiConv), expressed as:

$$g_i = g_i^L + g_i^{DH}, \tag{1}$$

where $g_i^L$ and $g_i^{DH}$ represent the outputs of LoConv and DyHiConv, respectively. LoConv captures basic geometric structures, while DyHiConv adaptively models fine-grained details through dynamic power functions.

#### 3.3.2 Low-order Convolution

LoConv primarily extracts basic geometric information from local structures. We adopt Nonparameter Trigonometric Functions (NTF) to encode point cloud coordinates $p_i$ and color information $c_i \in \mathbb{R}^3$, mapping them to the same dimension as their features, then adding their information and applying a non-linear transformation to obtain a high-dimensional representation of basic structural information. The LoConv can be formulated as:

$$g_i^L = \mathcal{A}(\{\mathbf{W}_l f_j' | p_j \in \mathcal{N}(p_i)\}), \tag{2}$$

$$f_j' = (f_j^p + f_j^c + f_j)/3, \tag{3}$$

$$f_j^p = [\sin(\alpha p_j / \beta^{\frac{6i}{d}}), \cos(\alpha p_j / \beta^{\frac{6i}{d}})]_{i=1}^d \in \mathbb{R}^d, \tag{4}$$

where $f_j^c$ is obtained similarly to $f_j^p$. $\alpha$ and $\beta$ represent the wavelength and amplitude hyperparameters of the trigonometric functions, respectively. $\mathbf{W}_l \in \mathbb{R}^{C_{in} \times C_{out}}$ denotes the non-linear transformation matrix.

#### 3.3.3 Dynamic High-order Convolution

To capture the details of complex local geometric structures, DyHiConv draws inspiration from dynamic convolution [46] to generate multiple convolution weights using input information. Unlike traditional dynamic convolution, we improve the power function design by using a smooth power function $(|f_j - f_i| + \epsilon)^{p_t}$ instead of traditional sign functions, where $p_t$ is a learnable parameter for each expert and $\epsilon$ is a small constant (typically $10^{-6}$). This design effectively captures local geometric details while maintaining gradient continuity. DyHiConv (see Fig. 2(b)) can be expressed as:

$$g_i^{DH} = \sum_{t=1}^T \phi_t g_i^t, \tag{5}$$

where $T$ is the number of experts (set to 8 in our implementation), $g_i^t$ represents the high-order convolution of the $t$-th expert, and $\phi_t$ represents the attention assembly coefficient of the expert. Specifically, the high-order features $g_i^t$ generated by each expert are calculated through:

$$g_i^t = \mathcal{A}(\{w_t(p_j) \odot (|f_j - f_i| + \epsilon)^{p_t} | p_j \in \mathcal{N}(p_i)\}), \tag{6}$$

where $w_t(p_j)$ is the dynamically generated weight, $p_t$ is the learnable power exponent parameter of the $t$-th expert, $\odot$ represents element-wise multiplication, and $\mathcal{A}$ is an aggregation function (typically max pooling).

The attention assembly coefficients $\phi_t$ are constructed from explicit geometric information $h_j$:

$$\phi_t = \frac{\exp(\mathbf{W}_t h_j)}{\sum_{t=1}^T \exp(\mathbf{W}_t h_j)}, \tag{7}$$

where $\mathbf{W}_t$ is a learnable transformation matrix.

### 3.3.4 Explicit Structure Introduction

To better utilize the geometric information of point clouds, we use the coordinates of neighboring points $p_j$ and center point $p_i$ as basic geometric elements to construct the weight $w_j$ for HiConv:

$$w_j = \mathbf{W}_h h_j, \tag{8}$$

where $h_j = [p_i, p_j, p_j - p_i, \|p_j - p_i\|] \in \mathbb{R}^{10}$, and $\mathbf{W}_h \in \mathbb{R}^{10 \times C_{out}}$ denotes the transformation matrix. The introduction of explicit geometric information facilitates the learning of relative spatial layout relationships between points and the capture of local geometric features and details.

### 3.4 Visual Introspective Prototype Module

To address feature discrepancies between support and query sets (e.g., intra-class diversity and domain gaps), we propose the Visual Introspective Prototype Module (VIP). It employs a multi-step reasoning process to iteratively refine prototypes, simulating human "think-reflect-revise" reasoning. VIP consists of two components: the Prototype Enhancement Module (PEM) and Prototype Difference Module (PDM), which alternately form a reasoning chain, progressively aligning feature distributions.

#### 3.4.1 Prototype Enhancement Module

The PEM aims to enhance the discriminability of prototype features and reduce intra-class diversity through self-attention and cross-attention mechanisms. Given a point cloud with $M$ points, let $\mathbf{F}_s \in \mathbb{R}^{M \times C}$ and $\mathbf{F}_q \in \mathbb{R}^{M \times C}$ denote the support and query features, respectively, where $C$ is the feature dimension. The PEM first applies local max pooling and projection mapping to extract statistical characteristics of each channel:

$$\mathbf{F}'_s = \text{MaxPool}(\mathbf{F}_s) \cdot \mathbf{W}_1, \qquad \mathbf{F}'_q = \text{MaxPool}(\mathbf{F}_q) \cdot \mathbf{W}_1, \tag{9}$$

where $\mathbf{W}_1 \in \mathbb{R}^{C \times C}$ is a learnable transformation matrix. Next, the PEM enhances the prototype features from two aspects:

1) **Self-correlation Enhancement**: The PEM learns internal structural information by computing self-correlation matrices of the support and query features:

$$\mathbf{A}_s = \mathbf{W}_3(\mathbf{F}'^T_s \mathbf{F}'_s), \qquad\qquad \mathbf{A}_q = \mathbf{W}_3(\mathbf{F}'^T_q \mathbf{F}'_q). \tag{10}$$

$$\mathbf{F}^{self}_p = \text{Softmax}(\mathbf{A}_s)\mathbf{F}_p + \text{Softmax}(\mathbf{A}_q)\mathbf{F}_p, \tag{11}$$

2) **Cross-correlation Enhancement**: The PEM learns shared information through interaction between the support and query features:

$$\mathbf{A}_{cross} = \mathbf{F}'^T_q \mathbf{F}'_s, \tag{12}$$

$$\mathbf{F}^{cross}_p = \text{Softmax}(\mathbf{A}_{cross}) \odot \mathbf{F}_p, \tag{13}$$

Finally, the enhanced prototype features output by the PEM are:

$$\mathbf{F}^e_p = \mathbf{F}^{self}_p + \mathbf{F}^{cross}_p + \mathbf{F}_p, \tag{14}$$

where $\mathbf{F}_p \in \mathbb{R}^{(K+1) \times C}$ represents the initial prototype features, and $\mathbf{F}^e_p$ denotes the enhanced prototype features.

#### 3.4.2 Prototype Difference Module

The PDM focuses on learning the differences between the support and query feature distributions to further eliminate domain gaps. After sharing similar pooling and mapping operations with the PEM, the PDM calculates the difference information between the support and query features:

$$\triangle_G = \mathbf{F}'^T_q \mathbf{F}'_q - \mathbf{F}'^T_s \mathbf{F}'_s, \tag{15}$$

and uses this difference information to adjust the prototype features:

$$\mathbf{F}^{delta}_p = \text{sigmoid}(\triangle_G) \odot \mathbf{F}^e_p. \tag{16}$$

Additionally, the PDM further optimizes the prototype features through cross-attention:

$$\mathbf{F}_p^{e\_cross} = \text{Softmax}(\mathbf{A}_{cross}) \odot \mathbf{F}_p^e, \tag{17}$$

where $\mathbf{A}_{cross}$ is the cross-correlation matrix between the support and query features. The final prototype features output by the PDM are:

$$\mathbf{F}_p^r = \mathbf{F}_p^{delta} + \mathbf{F}_p^{e\_cross} + \mathbf{F}_p^e, \tag{18}$$

where $\mathbf{F}_p^e$ is the output features from PEM, $\mathbf{F}_p^r$ is the output features after PDM processing.

### 3.4.3 Multi-step Reasoning Process

We design a multi-step reasoning process to progressively optimize the prototype features. Specifically, during $S$ reasoning steps, the PEM and PDM work alternately:

$$\mathbf{F}_p^t = \begin{cases} \text{PEM}(\mathbf{F}_q, \mathbf{F}_s, \mathbf{F}_p^{t-1}), & \text{if } t\%2 = 0 \\ \text{PDM}(\mathbf{F}_q, \mathbf{F}_s, \mathbf{F}_p^{t-1}) + \mathbf{F}_p^{t-1}, & \text{if } t\%2 = 1 \end{cases} \tag{19}$$

where $\mathbf{F}_p^t$ represents the prototype features at step $t$, and $\mathbf{F}_p^0$ represents the initial prototype features. This multi-step reasoning process enables iterative refinement: the PEM step enhances discriminability by extracting key information, while the PDM step eliminates domain gaps through difference learning, progressively aligning feature distributions.

### 3.4.4 Gating Fusion Network

To effectively fuse the multi-step reasoning results, we design a gating fusion network. At each reasoning step $t$, we compute the cosine similarity between the query features $\mathbf{F}_q$ and the prototype features $\mathbf{F}_p^t$. The intermediate prediction results $\mathbf{L}_t$ are then computed as:

$$\mathbf{L}_t = \text{sim}(\mathbf{F}_q, \mathbf{F}_p^t) \cdot \mathbf{L}_p = \frac{\mathbf{F}_q \cdot \mathbf{F}_p^t}{\|\mathbf{F}_q\|\|\mathbf{F}_p^t\|} \cdot \mathbf{L}_p, \tag{20}$$

where $\text{sim}(\cdot, \cdot)$ denotes the cosine similarity function that measures the semantic alignment between query and prototype features, $\cdot$ denotes the dot product, and $\|\cdot\|$ represents the L2 norm. $\mathbf{L}_p$ represents the prototype labels.

Finally, the gating network learns importance weights $\mathbf{w}$ for predictions from each step and fuses all predictions to obtain the final result. The final prediction $\mathbf{L}_{final}$ is obtained by a weighted sum:

$$\mathbf{L}_{final} = \sum_{t=1}^{S} \mathbf{w}_t \cdot \mathbf{L}_t = \sum_{t=1}^{S} \mathbf{GFN}(\mathbf{F_q}) \cdot \mathbf{L}_t, \tag{21}$$

where $\mathbf{w} \in \mathbb{R}^S$ is the weight from the gating network $\mathbf{GFN}(\cdot)$, and $\mathbf{L}_{final}$ is the final prediction.

## 4 Experiments

### 4.1 Datasets and Evaluation Metrics

We evaluate VIP-Seg on two widely adopted 3D segmentation benchmarks.

**S3DIS** dataset [3] consists of RGB point clouds collected from 272 rooms across 6 indoor areas. Each point is annotated with one of 13 semantic labels (12 object categories and clutter). Following the common practice [54], we split each scene into 1m × 1m blocks and sample 2,048 points per block, resulting in a total of 7,547 blocks.

**ScanNet** dataset [5] contains 1,513 scanned indoor scenes with dense point-wise annotations over 20 semantic categories (excluding unannotated areas). Using the same preprocessing pipeline, we generate 36,350 blocks, each containing 2,048 points.

**Evaluation Metric:** We employ the mean Intersection-over-Union (mIoU), a standard metric for point cloud segmentation, to assess model performance.

Table 1: **Few-shot Results (%) on S3DIS.** $S_i$ denotes the split $i$ is used for testing. *Avg* is the average mIoU across splits. The best results are shown in **bold**.

| Method | 2-Way | | | | | | 3-Way | | | | | |
|---|---|---|---|---|---|---|---|---|---|---|---|---|
| | 1-shot | | | 5-shot | | | 1-shot | | | 5-shot | | |
| | $S_0$ | $S_1$ | *Avg* | $S_0$ | $S_1$ | *Avg* | $S_0$ | $S_1$ | *Avg* | $S_0$ | $S_1$ | *Avg* |
| DGCNN [40] | 36.34 | 38.79 | 37.57 | 56.49 | 56.99 | 56.74 | 30.05 | 32.19 | 31.12 | 46.88 | 47.57 | 47.23 |
| ProtoNet [26] | 48.39 | 49.98 | 49.19 | 57.34 | 63.22 | 60.28 | 40.81 | 45.07 | 42.94 | 49.05 | 53.42 | 51.24 |
| MPTI [54] | 52.27 | 51.48 | 51.88 | 58.93 | 60.56 | 59.75 | 44.27 | 46.92 | 45.60 | 51.74 | 48.57 | 50.16 |
| AttMPTI [54] | 53.77 | 55.94 | 54.86 | 61.67 | 67.02 | 64.35 | 45.18 | 49.27 | 47.23 | 54.92 | 56.79 | 55.86 |
| BFG [19] | 55.60 | 55.98 | 55.79 | 63.71 | 66.62 | 65.17 | 46.18 | 48.36 | 47.27 | 55.05 | 57.80 | 56.43 |
| 2CBR [55] | 55.89 | 61.99 | 58.94 | 63.55 | 67.51 | 65.53 | 46.51 | 53.91 | 50.21 | 55.51 | 58.07 | 56.79 |
| PAP3D [9] | 59.45 | 66.08 | 62.76 | 65.40 | 70.30 | 67.85 | 48.99 | 56.57 | 52.78 | 61.27 | 60.81 | 61.04 |
| Seg-PN [56] | 64.84 | 67.98 | 66.41 | 67.63 | 71.48 | 69.56 | 59.11 | 60.42 | 59.77 | 59.48 | 64.72 | 62.10 |
| TaylorSeg-PN [38] | 67.12 | 71.11 | 69.12 | 70.44 | 72.23 | 71.34 | 60.28 | 65.70 | 63.00 | 62.78 | 67.06 | 64.33 |
| DAFNet [36] | 68.13 | 70.27 | 69.20 | 70.51 | 73.15 | 71.83 | 61.33 | 65.55 | 63.44 | 65.25 | 68.67 | 66.96 |
| DyPolySeg [35] | 72.02 | 73.82 | 72.92 | 75.99 | 75.32 | 75.66 | 64.54 | 67.93 | 66.24 | 65.61 | 70.22 | 67.92 |
| **VIP-Seg** | **72.20** | **76.09** | **74.15** | **76.48** | **77.54** | **77.01** | **68.80** | **68.35** | **68.58** | **68.15** | **70.44** | **69.30** |
| *Improvement* | +0.18 | +2.27 | +1.23 | +0.49 | +2.22 | +1.35 | +4.26 | +0.42 | +2.34 | +2.54 | +0.22 | +1.38 |

Table 2: **Few-shot Results (%) on ScanNet.** $S_i$ denotes the split $i$ is used for testing. *Avg* is the average mIoU across splits. The best results are shown in **bold**.

| Method | 2-Way | | | | | | 3-Way | | | | | |
|---|---|---|---|---|---|---|---|---|---|---|---|---|
| | 1-shot | | | 5-shot | | | 1-shot | | | 5-shot | | |
| | $S_0$ | $S_1$ | *Avg* | $S_0$ | $S_1$ | *Avg* | $S_0$ | $S_1$ | *Avg* | $S_0$ | $S_1$ | *Avg* |
| DGCNN [40] | 31.55 | 28.94 | 30.25 | 42.71 | 37.24 | 39.98 | 23.99 | 19.10 | 21.55 | 34.93 | 28.10 | 31.52 |
| ProtoNet [26] | 33.92 | 30.95 | 32.44 | 45.34 | 42.01 | 43.68 | 28.47 | 26.13 | 27.30 | 37.36 | 34.98 | 36.17 |
| MPTI [54] | 39.27 | 36.14 | 37.71 | 46.90 | 43.59 | 45.25 | 29.96 | 27.26 | 28.61 | 38.14 | 34.36 | 36.25 |
| AttMPTI [54] | 42.55 | 40.83 | 41.69 | 54.00 | 50.32 | 52.16 | 35.23 | 30.72 | 32.98 | 46.74 | 40.80 | 43.77 |
| BFG [19] | 42.15 | 40.52 | 41.34 | 51.23 | 49.39 | 50.31 | 34.12 | 31.98 | 33.05 | 46.25 | 41.38 | 43.82 |
| 2CBR [55] | 50.73 | 47.66 | 49.20 | 52.35 | 47.14 | 49.75 | 47.00 | 46.36 | 46.68 | 45.06 | 39.47 | 42.27 |
| PAP3D [9] | 57.08 | 55.94 | 56.51 | 64.55 | 59.64 | 62.10 | 55.27 | 55.60 | 55.44 | 59.02 | 53.16 | 56.09 |
| Seg-PN [56] | 63.15 | 64.32 | 63.74 | 67.08 | 69.05 | 68.07 | 61.80 | 65.34 | 63.57 | 62.94 | 68.26 | 65.60 |
| TaylorSeg-PN [38] | 67.52 | 70.75 | 69.14 | 68.39 | 71.55 | 69.97 | 63.60 | 67.55 | 65.58 | 66.98 | 69.78 | 68.38 |
| DAFNet [36] | 68.79 | 69.95 | 69.37 | 70.91 | 70.60 | 70.76 | 66.14 | 66.70 | 66.42 | 68.97 | 71.95 | 70.46 |
| DyPolySeg [35] | 71.05 | **72.73** | 71.89 | 71.25 | 73.66 | 72.46 | 67.65 | 71.24 | 69.45 | 68.73 | 69.62 | 69.18 |
| **VIP-Seg** | **71.59** | 72.64 | **72.12** | **72.35** | **72.90** | **72.63** | **68.87** | **71.38** | **70.13** | **70.84** | **71.72** | **71.28** |
| *Improvement* | +0.54 | -0.09 | +0.23 | +1.10 | -0.76 | +0.17 | +1.22 | +0.14 | +0.68 | +2.11 | +0.10 | +1.10 |

## 4.2 Comparison with Existing Methods

**Results analysis on the S3DIS dataset.** As shown in Table 1, our VIP-Seg demonstrates superior performance on the S3DIS dataset. In the 2-way 1-shot setting, VIP-Seg achieves an average mIoU of 74.15%, surpassing the previous best method DyPolySeg [35] by 1.23 percentage points. In the 2-way 5-shot setting, VIP-Seg reaches an average mIoU of 77.01%, exceeding DyPolySeg by 1.35 percentage points. In the 3-way settings, VIP-Seg achieves 68.58% and 69.30% mIoU in 1-shot and 5-shot scenarios respectively, outperforming DyPolySeg by 2.34 and 1.38 percentage points. These consistent gains across different settings validate the effectiveness and robustness of our approach. The improvements stem from our DyPowerConv's ability to capture local geometric features and the VIP module's effective prototype refinement through multi-step reasoning.

**Results analysis on the ScanNet dataset.** Our VIP-Seg also exhibits impressive performance on the more challenging ScanNet dataset, as illustrated in Table 2. In the 2-way 1-shot setting, VIP-Seg achieves an average mIoU of 72.12%, outperforming the previous best method DyPolySeg [35] by 0.23 percentage points. In the 2-way 5-shot setting, VIP-Seg attains an average mIoU of 72.63%, surpassing DyPolySeg by 0.17 percentage points. In the 3-way 1-shot setting, VIP-Seg achieves 70.13% mIoU, exceeding DyPolySeg by 0.68 percentage points. Particularly noteworthy is the 3-way 5-shot setting, where VIP-Seg achieves an average mIoU of 71.28%, outperforming DyPolySeg by 1.10 percentage points. These consistent improvements demonstrate VIP-Seg's ability to effectively handle complex indoor scenes with greater category diversity and varying levels of point cloud density. The multi-step reasoning mechanism proves particularly beneficial in distinguishing semantically

Table 3: Ablation study on the key components of VIP-Seg.

| LoConv | DyHiConv | Mamba | VIP | $S_0$ | $S_1$ | Avg |
|---|---|---|---|---|---|---|
| ✓ | ✗ | ✗ | ✗ | 48.92 | 51.15 | 50.04 |
| ✗ | ✓ | ✗ | ✗ | 49.87 | 52.34 | 51.11 |
| ✓ | ✓ | ✗ | ✗ | 51.98 | 54.02 | 53.00 |
| ✓ | ✓ | ✓ | ✗ | 53.64 | 56.21 | 54.93 |
| ✓ | ✓ | ✗ | ✓ | 70.55 | 71.34 | 70.95 |
| ✓ | ✓ | ✓ | ✓ | **72.20** | **76.09** | **74.15** |

Table 4: Ablation study on different components of the VIP module.

| PEM | PDM | 2-way-1-shot | | | 3-way-1-shot | | |
|---|---|---|---|---|---|---|---|
| | | $S_0$ | $S_1$ | Avg | $S_0$ | $S_1$ | Avg |
| ✗ | ✗ | 53.64 | 56.21 | 54.93 | 48.37 | 51.86 | 50.12 |
| ✓ | ✗ | 70.16 | 72.73 | 71.45 | 66.47 | 66.05 | 66.26 |
| ✗ | ✓ | 71.54 | 73.26 | 72.40 | 67.02 | 67.35 | 67.19 |
| ✓ | ✓ | **72.20** | **76.09** | **74.15** | **68.35** | **68.58** | **68.15** |

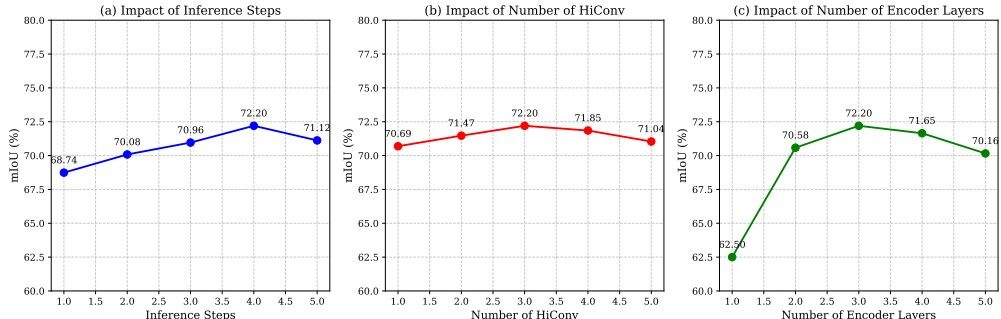

Figure 3: Parameter sensitivity analysis of VIP-Seg framework. (a) Impact of the number of reasoning steps in the VIP module on segmentation performance. (b) Impact of the number of HiConv experts in the DyPowerConv module. (c) Impact of encoder depth on model performance.

similar objects in cluttered environments, and the performance gains across all settings further validate the generalization capability of our approach for PC-FSS.

## 4.3 Ablation Experiments

All results are reported under 2-way-1-shot settings on the $S_0$ split of the S3DIS dataset.

### 4.3.1 Effectiveness of Different Components

Table 3 presents the ablation study on different components of our VIP-Seg. Using only LoConv or DyHiConv yields similar performance (50.04% and 51.11% mIoU), while their combination in the DyPowerConv module improves results to 53.00%. Adding Mamba blocks provides a modest gain of 1.93 percentage points, reaching 54.93% mIoU. Most significantly, incorporating the VIP module leads to a substantial improvement of 17.95 percentage points, increasing mIoU from 53.00% to 70.95% without Mamba blocks. The complete architecture achieves the best performance of 74.15% mIoU, confirming that each component contributes to the overall framework, with the VIP module providing the most significant impact.

### 4.3.2 Analysis of VIP Module Components

Table 4 shows the ablation study on different components of our VIP module. Without either PEM or PDM (baseline), the model achieves 54.93% and 50.12% mIoU on 2-way and 3-way settings, respectively. Using only PEM significantly improves performance to 71.45% (+16.52%) for 2-way and 66.26% (+16.14%) for 3-way settings, demonstrating its effectiveness in enhancing prototype discriminability. Similarly, using only PDM yields substantial improvements to 72.40% (+17.47%) for 2-way and 67.19% (+17.07%) for 3-way settings, indicating its ability to eliminate domain gaps. The complete VIP module with both PEM and PDM achieves the best performance across all settings, reaching 74.15% for 2-way and 68.15% for 3-way, validating the complementary effects of both components.

### 4.3.3 Hyperparameter Analysis

In Fig. 3(a), we analyze the effect of different reasoning steps in the VIP module. Performance improves from 1 step (68.74% mIoU) to 4 steps (72.20% mIoU), but decreases at 5 steps (71.12% mIoU),

Table 5: Impact of different geometric information in the explicit structure $h_j$ on VIP-Seg performance.

| Setting | 2-way-1-shot | | | 3-way-1-shot | | |
|---|---|---|---|---|---|---|
| | $S_0$ | $S_1$ | Avg | $S_0$ | $S_1$ | Avg |
| $[p_j]$ | 70.54 | 73.03 | 71.79 | 66.73 | 66.84 | 66.79 |
| $[p_i, p_j]$ | 71.67 | 74.25 | 72.96 | 67.36 | 67.05 | 67.21 |
| $[p_i, p_j, p_j - p_i]$ | 71.17 | 75.12 | 73.15 | 67.32 | 67.65 | 67.49 |
| $[p_i, p_j, p_j - p_i, \|p_i, p_j\|]$ | **72.20** | **76.09** | **74.15** | **68.35** | **68.58** | **68.15** |

Table 6: Performance and computational efficiency comparison.

| Method | mIoU | Param. | Pre-train Time | Episodic Train | Total Time |
|---|---|---|---|---|---|
| DGCNN [40] | 36.34 | 0.62 M | 4.0 h | 0.8 h | 4.8 h |
| AttMPTI [54] | 53.77 | 0.37 M | 4.0 h | 5.5 h | 9.5 h |
| 2CBR [55] | 55.89 | 0.35 M | 6.0 h | 0.2 h | 6.2 h |
| PAP3D [9] | 59.45 | 2.45 M | 3.6 h | 1.1 h | 4.7 h |
| Seg-PN [56] | 64.84 | 0.24 M | 0.0 h | 0.5 h | 0.5 h |
| VIP-Seg | **72.20** | 2.76 M | 0.0 h | 0.9 h | 0.9 h |

indicating optimal refinement occurs at 4 steps while excessive iterations cause over-processing. Fig. 3(b) shows the impact of HiConv layers. Performance increases from 1 layer (70.69% mIoU) to 3 layers (72.20% mIoU), then decreases with 4 layers (71.85% mIoU) and 5 layers (71.04% mIoU), suggesting 3 layers optimally balance feature capture and model complexity. Fig. 3(c) depicts the effect of encoder layers. The model improves dramatically from 1 layer (62.50% mIoU) to 3 layers (72.20% mIoU), gaining 9.70 percentage points, but declines with 4 layers (71.65% mIoU) and 5 layers (70.16% mIoU) due to overfitting. These results validate our architectural design choices.

### 4.3.4 Impact of Explicit Geometric Structure

Table 5 examines the influence of different geometric information in the explicit structure $h_j$ used in our DyPowerConv. Using only neighboring points $[p_j]$ achieves 71.79% and 66.79% mIoU on 2-way and 3-way settings, respectively. Adding center points $[p_i, p_j]$ improves performance to 72.96% (+1.17%) and 67.21% (+0.42%), demonstrating the importance of spatial context. Further incorporating relative displacement $[p_j - p_i]$ brings the performance to 73.15% (+0.19%) and 67.49% (+0.28%), capturing local geometric relationships more effectively. The complete representation that includes Euclidean distance $\|p_i - p_j\|$ achieves the best results of 74.15% and 68.15% mIoU across 2-way and 3-way settings, with improvements of 1.00% and 0.66% respectively, validating the complementary nature of different geometric features in enhancing the discriminative power of our DyPowerConv module.

### 4.4 Computational Complexity

Table 6 compares the computational efficiency of VIP-Seg with existing methods. Despite having 2.76M parameters, our approach eliminates pre-training, significantly reducing overall training time. Compared to pre-training methods like PAP3D [9] (4.7h total) and 2CBR [55] (6.2h total), VIP-Seg requires only 0.9h of training—an 81-85% reduction in total training time. When compared to Seg-PN [56], another pre-training-free method, VIP-Seg achieves a 7.36 percentage points higher mIoU (72.20% vs. 64.84%) with only 0.4h additional training time, demonstrating a superior balance between computational efficiency and performance. Notably, VIP-Seg achieves the highest performance among all methods while maintaining competitive training efficiency.

## 5 Conclusion

In this paper, we propose VIP-Seg, a novel pre-training-free framework for point cloud few-shot semantic segmentation that effectively addresses the challenges of intra-class diversity and domain gaps. Our approach introduces two key innovations: the VIP module, which employs a multi-step reasoning process to progressively refine prototype features, and DyPowerConv, which adaptively models local geometric structures through learnable power functions. Extensive experiments on S3DIS and ScanNet datasets demonstrate that VIP-Seg significantly outperforms current state-of-the-art methods across various few-shot settings. **Limitations:** While VIP-Seg eliminates the need for pre-training, the episode-based training paradigm still requires substantial computational resources, and performance may degrade with extremely sparse point clouds or significant occlusions. The optimal number of reasoning steps in the VIP module is dataset-dependent and may require tuning for different scenarios. **Future Work:** Incorporating multi-modal information (e.g., RGB images, depth maps) could enhance robustness to sparse or occluded point clouds, and exploring parameter-efficient integration of large-scale pre-trained models could potentially combine the benefits of pre-training with our efficient few-shot learning approach.

## Acknowledgments

This work was supported in part by the European Union's Horizon 2024 Research and Innovation Programme for the Marie Skłodowska-Curie Actions under Grant No. 101211118. This work was also supported by the UKRI Future Leaders Fellowship [MR/V025333/1] (RoboHike). Shuting He was sponsored by Shanghai Pujiang Programme 24PJD030 and Natural Science Foundation of Shanghai 25ZR1402138.

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
