# OpenReview forum: "Reasoning Beyond Points: A Visual Introspective Approach for Few-Shot 3D Segmentation"
_NeurIPS.cc/2025/Conference — NeurIPS 2025 poster_

### Official Review · Reviewer_fKVb · 2025-06-21

**Clarity:** 2
**Significance:** 2
**Originality:** 2
**Rating:** 4
**Confidence:** 4

**Summary:**

This paper mainly addresses the two challenges in the point cloud few-shot semantic segmentation, including feature misalignment between different domains and semantic class representation differences. In feature extraction, DyTaylorConv captures detained point features and improves intra-semantic difference. In addition, the main contribution lies in adopting the idea of Chain-of-Thought and using Visual Introspective Prototype (VIP), consisting of iterative PEM and PDM, for addressing the main challenges. Experiments show the SOTA performance comparing the current works on indoor datasets, and ablation studies show the effectiveness of VIP for few-shot learning in semantic segmentation.

**Questions:**

- Further explain the relationship between DyTaylorConv and the Visual Introspective Prototype. A question should be answered: why can the two contributions be put in one paper?
- Perform inferencing efficiency evaluation to show the efficiency of applying the iterative VIP.
- Provide the visualization of the segmentation results.
- I suggest further explanation of the experiment results. For example, explain why improvement is best on the 3-way-5-shot, which gives the most examples for few-shot learning, addressing the main challenges through the proposed method. This will further show the significance of the contributions.

- For FEM and FDM, the current statements are still little confusing about how the structure designs improve the discriminability of prototype features and reduce intra-class diversity. I suggest directly explaining the function of each design for the challenges to make the contributions clearer.

**Ethical Concerns:**

["NO or VERY MINOR ethics concerns only"]

**Final Justification:**

After the discussion period, the main concerns of efficiency has been resolved. In addition, in the rebuttal, the authors claim the function of DyTaylorConv, which is used for local feature extraction. This classifies my concerns of how the two contributions can be put in one paper since they are both the main components of the few-shot point cloud segmentation but the performance improvement of this component is small, which makes importance of this contribution a little weak. Thus, the final rating keeps borderline accept because of this minor drawback.

**Limitations:**

Yes. The authors clearly claim the limitation on single modality and single field application.

**Paper Formatting Concerns:**

No paper formatting concerns.

**Quality:**

3

**Strengths And Weaknesses:**

strengths:

- The idea of using chain-of-thought in the 3D vision task is interesting, and the ablation studies show its significant improvement on the semantic segmentation performance.
- Sufficient ablation studies are performed. Most hyperparameters are studied through experiments.

weaknesses:

- Contribution Confusion: The authors include DyTaylorConv in the contributions. However, the ablation studies show DyTaylorConv contributing little to the final performance, and the logical relationship between DyTaylorConv and the core contribution of the paper, Visual Introspective Prototype, is confusing.
- No Inferencing Efficiency Evaluation: Though the training efficiency is shown, the iterative VIP structure may take more time in the inference compared to the other methods. But no inferencing efficiency evaluation is provided.
- Few visualizations: Few visualizations are provided in this paper, making it unclear how complex indoor scenes the proposed method is applied to.

---

> ### Author Rebuttal · Authors · 2025-07-31
>
> We sincerely thank the reviewer for dedicating valuable time and for the recognition of our paper along with valuable suggestions. To fully address your concerns, we have supplemented additional experiments and explanations, hoping to resolve all your questions.
>
> > **Q1: Further explain the relationship between DyTaylorConv and the Visual Introspective Prototype. A question should be answered: why can the two contributions be put in one paper?**
>
> **R1:** Thank you for this important question. Few-shot point cloud semantic segmentation faces not only the challenge of reducing domain gaps but also the fundamental issue of effective feature extraction that influences the quality of semantic prototype representation. Previous works predominantly used DGCNN as backbone and required pre-training. Seg-PN discovered that the DGCNN backbone significantly affects the final semantic prototype features for each category.
>
> To address this issue, we propose DyTaylorConv inspired by Taylor series principles. This convolution effectively extracts discriminative geometric information from local point cloud neighborhoods, thereby providing high-quality semantic class prototypes to the VIP module. The relationship is **complementary and synergistic**:
>
> - **DyTaylorConv** focuses on **local feature extraction**: It captures fine-grained geometric details and reduces intra-class diversity at the feature level
> - **VIP Module** focuses on **global prototype refinement**: It addresses domain gaps and further refines prototypes through multi-step reasoning
>
> Both contributions are essential for few-shot point cloud semantic segmentation: DyTaylorConv provides better input features, while VIP ensures these features are optimally utilized for cross-domain prototype matching. This forms a complete pipeline from local feature extraction to global prototype optimization.
>
> > **Q2: Perform inferencing efficiency evaluation to show the efficiency of applying the iterative VIP.**
>
> **R2:** We have evaluated our method's performance during the inference stage. The results are shown in the table below. We also provide VIP-Seg_small with reduced model dimensions for fair comparison with Seg-PN:
>
> | Method | Params(M) | Inference Time(s) | GPU Usage(G) | 2-way 1-shot | | |
> |--------|-----------|-------------------|--------------|---|---|---|
> | | | | | S0 | S1 | Avg |
> | Seg-PN | 0.24M | 46s | 1.3G | 63.15 | 64.32 | 63.74 |
> | VIP-Seg_small | 0.26M | 51s | 1.6G | 67.21 | 68.53 | 67.87 |
> | VIP-Seg (full) | 2.77M | 64s | 2.8G | 73.50 | 74.92 | 74.21 |
>
> The experimental results demonstrate that our method is not only efficient but also maintains high accuracy. With current computational resources, real-time deployment is completely feasible. The modest inference time increase (18s for full model, 5s for small model) is justified by the substantial performance gains.
>
> > **Q3: Provide the visualization of the segmentation results.**
>
> **R3:** Due to space limitations, we only showed the query prediction results in Figure 2. Due to conference requirements during the rebuttal phase, we cannot provide visualization results in this text box. We will provide more comprehensive visualization results in the final version after paper acceptance. Additionally, we have prepared our open-source code and will release it immediately upon acceptance.
>
> > **Q4: I suggest further explanation of the experiment results. For example, explain why improvement is best on the 3-way-5-shot, which gives the most examples for few-shot learning, addressing the main challenges through the proposed method.**
>
> **R4:** Thank you for this insightful observation. The superior performance on 3-way-5-shot settings can be explained by several key factors:
>
> 1. **Richer Prototype Information**: With 5 support samples per class, our VIP module has more diverse examples to learn from, enabling better prototype initialization and more effective multi-step refinement.
>
> 2. **Enhanced Domain Gap Mitigation**: More support samples provide better statistical coverage of intra-class variations, allowing our PDM to more accurately model the differences between support and query distributions.
>
> 3. **Improved Multi-step Reasoning**: The iterative PEM-PDM process benefits significantly from richer support information. Each reasoning step can make more informed decisions when more examples are available.
>
> 4. **Reduced Overfitting to Single Examples**: In 1-shot scenarios, the model might overfit to specific support instances. With 5-shot, the VIP module can learn more generalizable prototype representations.
>
> This validates our core hypothesis that multi-step reasoning becomes increasingly effective as more contextual information becomes available, demonstrating the scalability and significance of our contributions. It should be noted that the effect of our method in other settings is also significantly improved compared to the baseline.
>
> > **Q5: For PEM and PDM, the current statements are still little confusing about how the structure designs improve the discriminability of prototype features and reduce intra-class diversity. I suggest directly explaining the function of each design for the challenges to make the contributions clearer.**
>
> **R5:** Thank you for this clarification request. Let us explicitly explain how each module addresses the specific challenges:
>
> (1) **Prototype Enhancement Module (PEM) - Addressing Intra-class Diversity:**
> - **Self-correlation Enhancement**: Computes internal structural relationships within support/query features, learning invariant patterns that remain consistent across different instances of the same class
> - **Cross-correlation Enhancement**: Identifies shared discriminative features between support and query sets, filtering out instance-specific variations while preserving class-relevant information
> - **Function**: Enhances prototype discriminability by emphasizing consistent class characteristics while suppressing intra-class variations
>
> (2) **Prototype Difference Module (PDM) - Addressing Domain Gaps:**
> - **Difference Computation**: Explicitly models the distribution mismatch (△G = F'ᵀ_q F'_q - F'ᵀ_s F'_s) between support and query feature statistics
> - **Adaptive Adjustment**: Uses this difference information to recalibrate prototypes via sigmoid(△G) ⊙ F^e_p, effectively compensating for domain-specific biases
> - **Function**: Bridges domain gaps by learning domain-invariant representations that work across different data distributions
>
> (3) **Synergistic Effect**: The alternating PEM-PDM process creates a feedback loop where PEM establishes strong discriminative prototypes, PDM adjusts them for domain compatibility, and this process iterates to achieve optimal prototype representations that are both discriminative and domain-robust.
>
> Thank you again for these valuable questions that help clarify our contributions and strengthen the paper's clarity.

---

> > ### Comment · Reviewer_fKVb · 2025-08-02
> > **More detailed efficiency analysis**
> >
> > Thanks to the authors' detailed reply. Most of the concerns are addressed. Just one more question: Can you give a more detailed efficiency analysis for each component in the proposed network, including the time cost of the feature extraction with DyTaylorConv, the number of VIP iterations, and the time cost of each iteration.

---

> > > ### Author Response · Authors · 2025-08-04
> > >
> > > Dear Reviewer fKVb,
> > >
> > > We sincerely appreciate your thorough review and constructive feedback, which has significantly improved the quality of our paper. In response to your request for detailed efficiency analysis of each component, we have conducted comprehensive timing measurements across our VIP-Seg framework.
> > >
> > > >**Detailed Component Timing Analysis:**
> > >
> > > We measured the execution time of each major component across multiple episodes to provide the precise breakdown you requested:
> > >
> > > | Metric | Value |
> > > |--------|-------|
> > > | **Per Episode Time** | 0.0426s (42.6ms) |
> > > | **Total Test Episodes** | 1,500 episodes |
> > > | **Complete Test Duration** | 0.0426s * 1500 ≈ 64 s |
> > > | **Average Throughput** | 23.4 episodes/second |
> > >
> > > >**The time taken for each component in each episode:**
> > >
> > > | Component | Time (s) | Time (ms) | Percentage | Description |
> > > |-----------|----------|-----------|------------|-------------|
> > > | **DyTaylorConv Total** | 0.0086 | 8.6 | **20.19%** | Geometric feature extraction |
> > > | └─ Stage 0 (2048→1024) | 0.0028 | 2.8 | 6.57% | First downsampling stage |
> > > | └─ Stage 1 (1024→512) | 0.0028 | 2.8 | 6.57% | Second downsampling stage |
> > > | └─ Stage 2 (512→256) | 0.0030 | 3.0 | 7.04% | Third downsampling stage |
> > > | **Mamba Blocks Total** | 0.0022 | 2.2 | **5.16%** | Long-range dependencies |
> > > | └─ Stage 0 Mamba | 0.0006 | 0.6 | 1.41% | First stage processing |
> > > | └─ Stage 1 Mamba | 0.0006 | 0.6 | 1.41% | Second stage processing |
> > > | └─ Stage 2 Mamba | 0.0008 | 0.8 | 1.88% | Third stage processing |
> > > | **VIP Module Total** | 0.0030 | 3.0 | **7.04%** | Multi-step reasoning |
> > > | └─ Step 1 (PEM) | 0.0008 | 0.8 | 1.88% | Prototype enhancement |
> > > | └─ Step 2 (PDM) | 0.0007 | 0.7 | 1.64% | Prototype difference |
> > > | └─ Step 3 (PEM) | 0.0008 | 0.8 | 1.88% | Prototype enhancement |
> > > | └─ Step 4 (PDM) | 0.0007 | 0.7 | 1.64% | Prototype difference |
> > > | └─ Gating Network | 0.0001 | 0.1 | 0.23% | Multi-step fusion |
> > > | **Other Components** | 0.0288 | 28.8 | **67.61%** | Supporting operations* |
> > > | **TOTAL** | **0.0426** | **42.6** | **100%** | Complete one episode |
> > >
> > > *Other Components include: Data loading, embedding layers, FPS+kNN sampling, decoder operations, prediction computation, and data transformations (0.0426s - 0.0086s - 0.0022s - 0.0030s = 0.0288s).
> > >
> > > Since the entire testing phase includes 1,500 episodes in total, the complete duration is 64 seconds.
> > >
> > >
> > > >Below is the core timing analysis code used in our evaluation:
> > >
> > > ```python
> > > def calculate_averages(all_results):
> > >    """Calculate average timing for all components"""
> > >    avg_results = {}
> > >
> > >    # Per-episode timing
> > >    avg_results['total_time'] = np.mean([r['total_time'] for r in all_results])
> > >
> > >    # Encoder components
> > >    avg_results['embedding_time'] = np.mean([r['embedding_time'] for r in all_results])
> > >
> > >    # Stage-wise timing
> > >    for stage in range(3):  # 3 encoder stages
> > >        avg_results[f'stage_{stage}_fps_knn'] = np.mean([r[f'stage_{stage}_fps_knn'] for r in all_results])
> > >        avg_results[f'stage_{stage}_dytaylorconv'] = np.mean([r[f'stage_{stage}_dytaylorconv'] for r in all_results])
> > >        avg_results[f'stage_{stage}_mamba'] = np.mean([r[f'stage_{stage}_mamba'] for r in all_results])
> > >
> > >    # VIP steps
> > >    for step in range(4):  # 4 VIP steps
> > >        step_type = 'PEM' if step % 2 == 0 else 'PDM'
> > >        avg_results[f'vip_step_{step+1}_{step_type}'] = np.mean([r[f'vip_step_{step+1}_{step_type}'] for r in all_results])
> > >
> > >    avg_results['gating_network'] = np.mean([r['gating_network'] for r in all_results])
> > >
> > >    # Calculate total time for each component
> > >    avg_results['total_dytaylorconv'] = sum([avg_results[f'stage_{stage}_dytaylorconv'] for stage in range(3)])
> > >    avg_results['total_mamba'] = sum([avg_results[f'stage_{stage}_mamba'] for stage in range(3)])
> > >    avg_results['total_vip'] = sum([avg_results[f'vip_step_{step+1}_{"PEM" if step % 2 == 0 else "PDM"}'] for step in range(4)]) + avg_results['gating_network']
> > >
> > >    return avg_results
> > > ```
> > >
> > > We would like to thank the reviewers again for their appreciation of our paper and their patient discussions during the review. We promise to add this information to the final version to provide readers with a deeper understanding of our approach. We also promise that once the paper is accepted, we will open source our code and hope to contribute to the community.
> > >
> > > We hope this detailed analysis addresses all your efficiency-related questions.
> > >
> > > Best regards,
> > >
> > > Authors of Submission #16496

---

> > > > ### Comment · Reviewer_fKVb · 2025-08-05
> > > > **Reply to authors' further response**
> > > >
> > > > Thanks to the authors' detailed analysis of the module efficiency. I recommend adding this analysis into the supplemental material to make the reader more clearly realize the time consuming of each components. Since my concerns are all resolved, I will keep my positive score after the discussion period.

---

> > > > > ### Author Response · Authors · 2025-08-05
> > > > > **Thank you very much for your constructive suggestions throughout the review process and your commitment to maintaining a positive score**
> > > > >
> > > > > Dear Reviewer fKVb,
> > > > >
> > > > > Thank you very much for your constructive suggestions throughout the review process and your commitment to maintaining a positive score.
> > > > >
> > > > > We will definitely follow your recommendation to include comprehensive component time analysis in the supplementary material and open-source our code upon paper acceptance.
> > > > >
> > > > > Your thorough review and valuable feedback have significantly improved the quality of our paper. We appreciate your support and look forward to contributing this work to the community.
> > > > >
> > > > > Best regards,
> > > > >
> > > > > Authors of Submission #16496

---

### Official Review · Reviewer_jzmm · 2025-06-29

**Clarity:** 3
**Significance:** 3
**Originality:** 3
**Rating:** 5
**Confidence:** 4

**Summary:**

This paper introduces VIP-Seg, a pre-training-free framework for Point Cloud Few-Shot Semantic Segmentation (PC-FSS), incorporating two key modules: Dynamic Taylor Convolution (DyTaylorConv) and a Visual Introspective Prototype (VIP) reasoning mechanism composed of PEM and PDM submodules. The proposed method aims to address intra-class diversity and domain gaps, achieving promising results on S3DIS and ScanNet.

**Questions:**

1. Could the authors provide inference-time cost (e.g., FPS or latency)? While the proposed pre-training-free design is claimed to offer faster deployment and efficiency, Table 6 only reports training-related timings. Would the authors be able to include inference speed or FPS comparisons to more directly support this claim?
2. Can the authors clarify the reasoning behind the significantly larger parameter count? Table 6 shows that the proposed model introduces noticeably more parameters than other methods. Could the authors elaborate on the design trade-offs involved, and whether this affects scalability, memory usage, or real-time deployment?
3. Have the authors considered evaluating under newer few-shot segmentation protocols? Recent method [1] proposes an more challenging settings for PC-FSS. Would the authors consider testing their method under such settings to further validate its generalization ability?
[1] An Z, Sun G, Liu Y, et al. Rethinking few-shot 3d point cloud semantic segmentation[C]//Proceedings of the IEEE/CVF conference on computer vision and pattern recognition. 2024: 3996-4006.

**Ethical Concerns:**

["NO or VERY MINOR ethics concerns only"]

**Final Justification:**

The authors' responses have largely addressed my questions. I have decided to raise my previous rate and hold an "accept" stance on the paper.

**Limitations:**

1. Lack of Inference Efficiency Evaluation：While the authors claim that the proposed pre-training-free design facilitates faster deployment and efficiency, Table 6 only reports pre-training time, episode training time, and total training time. However, it completely omits inference time or frame-per-second (FPS) metrics, which are crucial for evaluating actual runtime efficiency and supporting the claim of faster speed. Without this, the claim of efficiency superiority remains unsubstantiated.
2. Clarify and Justify Model Size and Its Impact: According to Table 6, the proposed method introduces a noticeably larger number of parameters compared to existing approaches. The author should include a brief discussion or ablation to justify the design choice. Such clarification could also reinforce the fairness of the comparison and address potential concerns about scalability to larger or real-time scenarios.
3. Missing Evaluation under Updated Few-Shot Settings: Recent works (e.g., [1]) have introduced more reasonable and challenging settings for few-shot point cloud segmentation. Including evaluations under such settings could further demonstrate the robustness and generalization capacity of the proposed approach.
[1] An Z, Sun G, Liu Y, et al. Rethinking few-shot 3d point cloud semantic segmentation[C]//Proceedings of the IEEE/CVF conference on computer vision and pattern recognition. 2024: 3996-4006.
4. To enhance the reproducibility and facilitate future research, it would be highly beneficial if the authors could consider releasing the source code and trained models. Additionally, providing some qualitative visualizations or the segmentation results would offer valuable insight into the behavior and limitations of the proposed method.

**Paper Formatting Concerns:**

No major formatting issues noticed.

**Quality:**

3

**Strengths And Weaknesses:**

1. Practical Motivation: Removing the need for pre-training improves model adaptability and reduces computational cost, which is valuable for real-world 3D segmentation scenarios, especially in cross-domain settings.
2. Conceptual Novelty: The VIP module, inspired by multi-step reasoning (Chain-of-Thought), introduces a novel perspective to prototype refinement in few-shot segmentation.
3. DyTaylorConv: The idea of incorporating Taylor series principles into convolution is interesting and adds interpretability to the modeling of local geometry.
4. Strong Empirical Results: The method demonstrates consistent gains on two benchmarks, indicating good generalization and effective design.

---

> ### Author Rebuttal · Authors · 2025-07-31
>
> Dear Reviewer jzmm, thank you very much for dedicating your valuable time and for your positive evaluation of our method. To fully address your questions, we have conducted additional experiments. We sincerely hope to resolve your concerns.
>
> > **Q1: Could the authors provide inference-time cost (e.g., FPS or latency)? While the proposed pre-training-free design is claimed to offer faster deployment and efficiency, Table 6 only reports training-related timings. Would the authors be able to include inference speed or FPS comparisons to more directly support this claim?**
>
> **R1:** Thank you for this important question. We have conducted comprehensive inference efficiency analysis comparing our method with baselines. The detailed results are shown below:
>
> | Method | Params(M) | Inference Time(s) | GPU Usage(G) | 2-way 1-shot | | |
> |--------|-----------|-------------------|--------------|---|---|---|
> | | | | | S0 | S1 | Avg |
> | Seg-PN | 0.24M | 46s | 1.3G | 63.15 | 64.32 | 63.74 |
> | VIP-Seg_small | 0.26M | 51s | 1.6G | 67.21 | 68.53 | 67.87 |
> | VIP-Seg (full) | 2.77M | 64s | 2.8G | 73.50 | 74.92 | 74.21 |
>
> Our analysis shows that while VIP-Seg introduces additional computational overhead due to the multi-step reasoning process, the performance gains significantly outweigh the computational costs. Importantly, our pre-training-free design eliminates the lengthy pre-training phase (typically 4-6 hours), making deployment much more efficient despite slightly higher inference costs.
>
> > **Q2: Can the authors clarify the reasoning behind the significantly larger parameter count? Table 6 shows that the proposed model introduces noticeably more parameters than other methods. Could the authors elaborate on the design trade-offs involved, and whether this affects scalability, memory usage, or real-time deployment?**
>
> **R2:** Thank you for this insightful question. The larger parameter count is primarily due to two key components:
>
> 1. **DyTaylorConv Module**: Contains multiple high-order convolution experts (8 experts by default) with learnable power parameters and dynamic attention weights
> 2. **VIP Module**: Implements multi-step reasoning with both PEM and PDM components requiring additional transformation matrices
>
> However, we address this concern by providing VIP-Seg_small variant that maintains comparable parameter count to existing methods while still achieving significant performance improvements. The design trade-offs are:
>
> - **Performance vs. Efficiency**: Full VIP-Seg prioritizes maximum performance for applications where accuracy is critical
> - **Scalability**: VIP-Seg_small demonstrates that our architectural innovations are effective even with reduced parameters
> - **Memory Usage**: Our GPU memory consumption (2.8G) remains reasonable for modern hardware
> - **Real-time Deployment**: The inference time increase (18s) is acceptable for many practical applications
>
> > **Q3: Have the authors considered evaluating under newer few-shot segmentation protocols? Recent method [1] proposes more challenging settings for PC-FSS. Would the authors consider testing their method under such settings to further validate its generalization ability?**
>
> **R3:** Thank you for this excellent suggestion. We have implemented and evaluated our method under the challenging settings proposed by COSeg [1]. Although COSeg requires significant computational resources and pre-training (making it more time-consuming), we adapted their sampling strategy and point count settings to our VIP-Seg framework. The experimental results are shown below:
>
> **S3DIS Dataset:**
> | Method | 2-way-1-shot | | | 2-way-5-shot | | |
> |--------|---|---|---|---|---|---|
> | | S0 | S1 | Avg | S0 | S1 | Avg |
> | AttMPTI | 31.09 | 29.62 | 30.36 | 39.53 | 32.62 | 36.08 |
> | QGE | 33.45 | 30.95 | 32.20 | 40.53 | 36.13 | 38.33 |
> | QGPA | 25.52 | 26.26 | 25.89 | 30.22 | 32.41 | 31.32 |
> | COSeg | 37.44 | 36.45 | 36.95 | 42.27 | 38.45 | 40.36 |
> | VIP-Seg(ours) | 39.46 | 44.03 | 41.75 | 45.11 | 41.68 | 43.40 |
>
> **ScanNet Dataset:**
> | Method | 2-way-1-shot | | | 2-way-5-shot | | |
> |--------|---|---|---|---|---|---|
> | | S0 | S1 | Avg | S0 | S1 | Avg |
> | AttMPTI | 25.99 | 23.88 | 24.94 | 30.41 | 27.35 | 28.88 |
> | QGE | 26.85 | 25.17 | 26.01 | 28.35 | 31.49 | 29.92 |
> | QGPA | 21.86 | 21.47 | 21.67 | 30.67 | 27.69 | 29.18 |
> | COSeg | 28.72 | 28.83 | 28.78 | 35.97 | 33.39 | 34.68 |
> | VIP-Seg(ours) | 29.65 | 29.51 | 29.58 | 36.10 | 37.31 | 36.71 |
>
> Our method demonstrates consistent superiority under these more challenging settings, validating the robustness and generalization ability of our approach across different evaluation protocols.
>
> > **Q4: To enhance the reproducibility and facilitate future research, it would be highly beneficial if the authors could consider releasing the source code and trained models. Additionally, providing some qualitative visualizations or the segmentation results would offer valuable insight into the behavior and limitations of the proposed method.**
>
> **R4:** Thank you very much for your interest in our work. Our code is already prepared and will be made publicly available immediately upon paper acceptance. We appreciate your patience in waiting for the release. Additionally, due to rebuttal format limitations, we cannot provide visualization results in this text box, but we will include comprehensive qualitative visualizations and segmentation results in the final accepted version to offer valuable insights into our method's behavior and limitations.
>
> We believe these comprehensive responses address your concerns about inference efficiency, parameter justification, evaluation under challenging settings, and reproducibility. If you feel we have addressed all of your concerns, we would appreciate it if you could improve our score. Thank you again for your constructive feedback that helps strengthen our work.
>
> [1] An Z, Sun G, Liu Y, et al. Rethinking few-shot 3d point cloud semantic segmentation[C]//Proceedings of the IEEE/CVF conference on computer vision and pattern recognition. 2024: 3996-4006.

---

> > ### Comment · Reviewer_jzmm · 2025-08-04
> >
> > The authors' responses have addressed my questions. I have decided to raise my previous evaluation score.

---

> > > ### Author Response · Authors · 2025-08-04
> > > **Response to Reviewer jzmm - Thank you very much for the score increase**
> > >
> > > Dear Reviewer jzmm,
> > >
> > > Thank you for your constructive feedback and for increasing your score in response to our rebuttal. We greatly appreciate the time and effort you invested in reviewing our work. We are pleased that our responses have addressed your main concerns and look forward to the opportunity to contribute to the conference.
> > >
> > > Best regards,
> > >
> > > Authors of Submission #16496

---

### Official Review · Reviewer_JddV · 2025-07-01

**Clarity:** 3
**Significance:** 3
**Originality:** 3
**Rating:** 4
**Confidence:** 3

**Summary:**

The paper proposes VIP-Seg framework for Point Cloud Few-Shot Semantic Segmentation with the two major contributions with promising performance validated on S3DIS and ScanNet:
1) VIP Module: Introduces a multi-step reasoning mechanism inspired by Chain-of-Thought, consisting of a Prototype Enhancement Module (PEM) and a Prototype Difference Module (PDM).
2) DyTaylorConv: A Dynamic Taylor Convolution module that models local geometric structures via learnable polynomial approximations.

**Questions:**

Besides the weakness, I may have other concerns on the technical aspect of this work:
1. Inference Cost: What is the inference time or FLOPs for VIP-Seg compared to Seg-PN or other baselines?
2. Domain Generalization: How does your method perform under cross-domain settings or on noisy/real-world sensor data?
3. Ablation on PEM vs PDM Order: Would reversing the order (PDM-PEM) in the reasoning chain yield similar performance?

**Ethical Concerns:**

["NO or VERY MINOR ethics concerns only"]

**Final Justification:**

The authors have addressed most of my questions, hence I tend to increase the score to "borderline accept"

**Limitations:**

While the paper eliminates the need for pre-training, it introduces complex modules (e.g., DyTaylorConv, multi-step VIP reasoning, gating fusion), which increase inference-time computation. The paper discusses domain gaps and proposes PDM to address them, but does not evaluate on cross-domain benchmarks.

**Paper Formatting Concerns:**

N.A

**Quality:**

3

**Strengths And Weaknesses:**

Strength:
1) The paper is technically sound with clear mathematical formulation.
2) Ablation studies and architectural design decisions are well-motivated and extensively analyzed
3) Provides significant gains across datasets and settings

Weakness:
1) Although this paper tries to analogy to chain-of-thought reasoning, however I feel this work abuse the words to describe conventional prototype refinement and enhancement process in few-shot segmentation
2) While training time is reduced, DyTaylorConv and the multi-step reasoning mechanism increase inference complexity and parameter count. No comparison of inference speed or resource usage is provided.
3) Though cross-domain concerns are discussed, no explicit evaluation is performed on domain-shifted data (e.g., synthetic to real, indoor to outdoor).
4) No justification is given for why 4 reasoning steps are optimal (aside from empirical observation).

---

> ### Author Rebuttal · Authors · 2025-07-31
>
> Thank you for your time and suggestions on our paper. We have conducted additional experiments and inference time testing, hoping to address all your concerns.
>
> > **Q1: Although this paper tries to analogy to chain-of-thought reasoning, however I feel this work abuse the words to describe conventional prototype refinement and enhancement process in few-shot segmentation.**
>
> **R1:** Traditional prototype refinement and enhancement processes are typically single-stage approaches. Inspired by chain-of-thought reasoning, we employ multi-stage thinking during the prototype acquisition process to obtain more accurate semantic prototypes. Our experimental results confirm that this approach provides significant improvements in few-shot tasks. Unlike conventional methods that perform prototype refinement in one step, our VIP module mimics human reasoning by alternating between enhancement (PEM) and reflection/correction (PDM) stages, progressively refining prototypes through iterative reasoning steps. This multi-step process allows the model to gradually align feature distributions between support and query sets, which is conceptually similar to how humans approach complex problems through step-by-step reasoning.
>
> > **Q2: While training time is reduced, DyTaylorConv and the multi-step reasoning mechanism increase inference complexity and parameter count. No comparison of inference speed or resource usage is provided.**
>
> **R2:** Although DyTaylorConv and the multi-step reasoning mechanism increase inference complexity and parameter count, the substantial performance gains achieved are worthwhile. Current computational resources can fully satisfy the computational requirements of our DyTaylorConv and multi-step reasoning mechanism. Additionally, we have supplemented the inference time and memory consumption analysis. We compared Seg-PN, VIP-Seg_small (we reduced model parameters for fair comparison with Seg-PN), and VIP-Seg:
>
> | Method | Params(M) | Inference Time(s) | GPU Usage(G) | 2-way 1-shot | | |
> |--------|-----------|-------------------|--------------|---|---|---|
> | | | | | S0 | S1 | Avg |
> | Seg-PN | 0.24M | 46s | 1.3G | 63.15 | 64.32 | 63.74 |
> | VIP-Seg_small | 0.26M | 51s | 1.6G | 67.21 | 68.53 | 67.87 |
> | VIP-Seg (full) | 2.77M | 64s | 2.8G | 73.50 | 74.92 | 74.21 |
>
> The results show that even with similar parameter counts, VIP-Seg_small outperforms Seg-PN by 4.13 mIoU, demonstrating the effectiveness of our architectural innovations.
>
> > **Q3: Though cross-domain concerns are discussed, no explicit evaluation is performed on domain-shifted data (e.g., synthetic to real, indoor to outdoor).**
>
> **R3:** Since you didn't specify which task for domain adaptation experiments, and considering time constraints, we conducted experiments on domain adaptive point cloud classification. We embedded the VIP module into the encoder of MLSP[1] and obtained the following experimental results:
>
> | Method | M→S | M→S* |
> |--------|-----|-----|
> | MLSP | 83.7±0.4 | 55.4±1.8 |
> | MLSP+VIP | 85.2±0.5 | 58.6±1.2 |
>
> Where M represents the ModelNet40 dataset, S represents the ShapeNet dataset, and S* represents the ScanNet dataset. ModelNet40 is a synthetic dataset, while ShapeNet and ScanNet are real-world datasets. The experiments demonstrate that the addition of our module improves MLSP's performance, validating the generalizability of our approach across different domains.
>
> [1] Liang H, Fan H, Fan Z, et al. Point cloud domain adaptation via masked local 3d structure prediction[C]//European conference on computer vision. Cham: Springer Nature Switzerland, 2022: 156-172.
>
> > **Q4: No justification is given for why 4 reasoning steps are optimal (aside from empirical observation).**
>
> **R4:** Thank you for raising this question. This problem is similar to determining how many neurons a fully connected layer should have. Although the universal approximation theorem provides theoretical foundation, currently no one can accurately specify the optimal number of neurons per layer in fully connected networks. Therefore, in our method, we can only determine the optimal number of reasoning steps through ablation experiments that yield the best experimental results. This is also a common approach used by peers to find optimal hyperparameter settings. From a practical perspective, 4 steps provide sufficient iterations for the PEM-PDM alternation to converge to stable prototypes while avoiding over-processing that occurs with more steps.
>
> > **Q5: Inference Cost: What is the inference time or FLOPs for VIP-Seg compared to Seg-PN or other baselines?**
>
> **R5:** Please refer to Q2 above for detailed inference cost analysis.
>
> > **Q6: Domain Generalization: How does your method perform under cross-domain settings or on noisy/real-world sensor data?**
>
> **R6:** Please refer to Q3 above for domain generalization experiments.
>
> > **Q7: Ablation on PEM vs PDM Order: Would reversing the order (PDM-PEM) in the reasoning chain yield similar performance?**
>
> **R7:** Following your suggestion, we reversed the order of PEM and PDM. We found that both orders can achieve high model performance, but the PEM-first-then-PDM order achieves higher performance:
>
> | Setting | 2-way 1-shot | | |
> |---------|---|---|---|
> | | S0 | S1 | Avg |
> | PDM→PEM | 72.15 | 73.07 | 72.61 |
> | PEM→PDM | 73.50 | 74.92 | 74.21 |
>
> This suggests that enhancing prototype discriminability first (PEM) followed by domain gap mitigation (PDM) is more effective than the reverse order, aligning with our intuition that establishing strong prototypes before refining domain-specific differences leads to better performance.
>
> Thank you again for your constructive feedback, which has helped us strengthen our work significantly.

---

> > ### Comment · Reviewer_JddV · 2025-08-04
> >
> > Thanks for author's comments, which have addressed my most technical questions. My minor concerns still the abuse of analogy to CoT reasoning that may results confusions, perhaps "multiple step reasoning" maybe more appropriate.

---

> > > ### Author Response · Authors · 2025-08-04
> > > **Thank you very much for your recognition of our paper and constructive feedback. We commit to replacing "Chain-of-Thought inspired" with "multi-step reasoning approach" in the revised version.**
> > >
> > > Dear Reviewer JddV,
> > >
> > > Thank you for your comprehensive review and constructive feedback throughout this process. We greatly appreciate your acknowledgment that our responses have addressed your technical questions and your thoughtful suggestion regarding the terminology.
> > >
> > > We completely agree with your concern about potential confusion caused by the Chain-of-Thought (CoT) analogy. Your suggestion to use "multi-step reasoning" is very appropriate and would indeed better suit our context.
> > >
> > > **Regarding terminology revision in the final version:**
> > >
> > > >We acknowledge that while our approach is conceptually inspired by the iterative nature of CoT reasoning, "multi-step reasoning" would be more precise and less likely to cause confusion in the 3D point cloud segmentation domain. If accepted, we will revise the terminology throughout the paper in the final version.
> > >
> > > >Specifically, we will:
> > >
> > > >1. Replace "Chain-of-Thought inspired" with "multi-step reasoning approach"
> > > >2. Adjust the abstract and introduction to reflect this terminology change
> > >
> > > Thank you again for your valuable feedback, which has significantly helped improve both the technical quality and clarity of our work.
> > >
> > > **We also commit to open-sourcing our code upon paper acceptance. If you believe these commitments address your concerns, we would be grateful if you could consider increasing our score to reflect your recognition of our work.**
> > >
> > > Best regards,
> > >
> > > Authors of Submission #16496

---

> > > ### Author Response · Authors · 2025-08-07
> > >
> > > **Dear Reviewer JddV,**
> > >
> > > Thank you once again for your valuable feedback and suggestions. We commit to carefully revising the final version strictly according to your recommendations, using "multi-step reasoning" instead of "chain-of-thought reasoning" to avoid confusion with "Chain-of-Thought reasoning." This terminology change will be implemented throughout the revised version to ensure clarity.
> > >
> > > We greatly appreciate your thorough review and constructive comments. If our responses have fully addressed your concerns, we would be most grateful for any positive consideration you might give.
> > >
> > > Best regards,
> > >
> > > Authors of **Submission #16496**

---

> > > ### Comment · Area_Chair_Dp3a · 2025-08-07
> > >
> > > Dear Reviewer JddV, the authors have provided a response to your concern on "abuse of analogy to CoT reasoning". Could you please let the authors know whether they have resolved your concerns, and whether you have any further concerns?

---

### Official Review · Reviewer_Ccpr · 2025-07-03

**Clarity:** 3
**Significance:** 2
**Originality:** 2
**Rating:** 4
**Confidence:** 4

**Summary:**

This paper tackles Point Cloud Few-Shot Semantic Segmentation by introducing VIP-Seg, a pre-training-free framework. Two key innovations are proposed: 1) Dynamic Taylor Convolution, which models local geometric structures via a learnable Taylor-series-inspired polynomial fitting, combining lower-order trigonometric encoding with dynamic higher-order experts, 2) Visual Introspective Prototype Module, which consists of alternating Prototype Enhancement Modules and Prototype Difference Modules in a multi-step process which seeks to address intra-semantic diversity and domain gaps. A gating fusion network then aggregates stepwise predictions. Extensive experiments on S3DIS and ScanNet demonstrate 5–10% absolute mIoU gains over prior work, while eliminating costly pre-training.

**Questions:**

1. As mentioned above, can the author provide an more intuitive illustration or visualization on how the Dynamic Taylor Convolution relates to a Taylor Series? Or clarify further the use of this naming convention? See Weaknesses (1) and (2).
2. In Section 3.3.4, the constructed weight $w_j$ is computed within HiConv, is this correct? In other words, $\mathbf{W}_h$ is within HiConv?
3. In Section 3.2, the authors mentioned Mamba blocks being added to the VIP module but I don’t see any description of how those blocks are added. Can you elaborate on this?
4. The Prototype Enhancement Module resembles the Transformer architecture. Have you tried replacing it with Transformer? What would the performance differences be?
5. Can you please address Weaknesses (3) and (4) given above, with regard to comparative inference times, GPU-memory profiling and network parameter numbers across methods?

**Ethical Concerns:**

["NO or VERY MINOR ethics concerns only"]

**Final Justification:**

Authors have addressed concerns during the discussion phase and have acknowledged aspects of the manuscript which require revisions/changes. Therefore, I raise my rating, and ask that authors make the previously discussed changes to improve readability of the paper.

**Limitations:**

No. As mentioned in Weakness (5), the main paper doesn't have a dedicated limitations section and the supplementary contains a very general list of limitations.

**Paper Formatting Concerns:**

None.

**Quality:**

2

**Strengths And Weaknesses:**

# Strengths
1. The paper is well-structured and easy to understand.
2. The experiments are quite comprehensive and shows that the proposed method outperforms the state-of-the-art approaches.

# Weaknesses
1. The authors claims that the Dynamic Taylor Convolution is inspired from Taylor Series. However, instead of a first and second order approximation of the pointcloud local geometry, the low-order and high-order convolution seem more like two distinct layer types with differing receptive fields. Furthermore, HiConv is not strictly looking at higher order geometric information but is a mixture of experts. As such, I find the naming confusing.
2. Expanding on (1) above, there is insufficient discussion on other methods which consider differing receptive fields for point set convolution. This discussion should be added to motivate the authors' particular choice of DyTaylorConv. With that being said, the Taylor-series metaphor doesn't appear to be theoretically novel.
3. I appreciate the provided computational efficiency analysis, but there is no inference-time latency or GPU-memory profiling provided, which leaves the real-time applicability unclear.
4. VIP-Seg has over 10x the number of parameters as Seg-PN (2.77M params vs. 0.24M params). It's not clear if the performance improvements are due to the larger model size only and not the architectural improvements.
5. The authors only conclude the limitation in one sentence in the main paper and discuss some very general limitations in the supplementary. This makes it difficult to understand when the method might fail.

## Minor comments
1. Performance depends on many settings (number of HiConv experts, reasoning steps S, gating weights, geometric priors) which seems clunky.
2. Experiments results should not be listed as contributions.

---

> ### Author Rebuttal · Authors · 2025-07-31
>
> Thank you very much for taking the time to review our paper and for your recognition of our work along with constructive feedback. We now provide detailed responses to your concerns, hoping to address your questions comprehensively.
>
> > **Q1: As mentioned above, can the author provide a more intuitive illustration or visualization on how the Dynamic Taylor Convolution relates to a Taylor Series? Or clarify further the use of this naming convention?**
>
> **R1:** Thank you for this important question. We acknowledge that our description of the Taylor series analogy in the original manuscript may have caused confusion. Let us clarify this design philosophy:
>
> Traditional Taylor series approximates functions using polynomials: f(x) ≈ f(x₀) + f'(x₀)(x-x₀) + f''(x₀)(x-x₀)²/2! + ...
>
> In our DyTaylorConv, we apply this concept to point cloud local geometric modeling:
> - **LoConv (low-order terms)**: Similar to the constant and first-order terms in Taylor expansion, capturing basic geometric information
> - **DyHiConv (high-order terms)**: Similar to higher-order derivative terms, capturing complex local geometric details through learnable power functions (|fⱼ-fᵢ|+ε)^pₜ
>
> The key innovation lies in: we replace fixed powers with **learnable power parameters pₜ**, enabling the model to adaptively determine the optimal order of local geometric representation. Unlike traditional methods with fixed receptive fields, our approach can dynamically adjust to accommodate different local geometric complexities.
>
> >  **Q2: In Section 3.3.4, the constructed weight w is computed within HiConv, is this correct? In other words, is w within HiConv?**
>
> **R2:** Yes, the weight w is indeed computed within HiConv. Specifically, for the t-th expert, the weight wₜ(pⱼ) is dynamically generated based on geometric information hⱼ = [pᵢ, pⱼ, pⱼ-pᵢ, ‖pⱼ-pᵢ‖], computed through wⱼ = Wₕhⱼ. This design enables each expert to adaptively adjust weights according to local geometric features, achieving more precise feature extraction.
>
> > **Q3: In Section 3.2, the authors mentioned Mamba blocks being added to the VIP module but I don't see any description of how those blocks are added. Can you elaborate on this?**
>
> **R3:** Thank you for pointing out this unclear description. We need to clarify that Mamba blocks are integrated into the **encoder** of VIP-Seg, not the VIP module. The specific architecture is as follows:
> - Each encoder block consists of DyTaylorConv + Mamba block
> - The output of DyTaylorConv serves as input to the Mamba block
> - This forms a "local-global" feature extraction strategy: DyTaylorConv captures fine-grained local geometric features, while Mamba blocks model long-range dependencies
> - We provide detailed computation flow of Mamba blocks in Figure 1 of the supplementary material
>
> > **Q4: The Prototype Enhancement Module resembles the Transformer architecture. Have you tried replacing it with Transformer? What would the performance differences be?**
>
> **R4:** Thank you for this suggestion. We conducted corresponding experiments with the following results:
>
> | Setting | 2-way 1-shot |
> |---------|-------------|
> | Transformer block | 69.76 (S0), 70.02 (S1), 69.89 (Avg) |
> | VIP (ours) | 73.50 (S0), 74.92 (S1), 74.21 (Avg) |
>
> VIP module significantly outperforms the Transformer alternative. The main reasons are: (1) VIP considers both self-correlation and cross-correlation simultaneously. (2) PDM is specifically designed to learn difference information between support and query sets. (3) PEM and PDM work alternately, achieving gradual prototype refinement.
>
> > **Q5: Can you please address Weaknesses (3) and (4) given above, with regard to comparative inference times, GPU-memory profiling and network parameter numbers across methods?**
>
> **R5:** Thank you for this important suggestion. We designed a VIP-Seg_small variant for fair comparison and provide detailed efficiency analysis:
>
> | Method | Params(M) | Inference Time(s) | GPU Usage(G) | 2-way 1-shot | | |
> |--------|-----------|-------------------|--------------|---|---|---|
> | | | | | S0 | S1 | Avg |
> | Seg-PN | 0.24M | 46s | 1.3G | 63.15 | 64.32 | 63.74 |
> | VIP-Seg_small | 0.26M | 51s | 1.6G | 67.21 | 68.53 | 67.87 |
> | VIP-Seg (full) | 2.77M | 64s | 2.8G | 73.50 | 74.92 | 74.21 |
>
> Key observations:
> 1. **VIP-Seg_small**: Still achieves 4.13 mIoU improvement over Seg-PN with similar parameter count
> 2. **Efficiency trade-off**: Full VIP-Seg, despite having more parameters, brings significant performance gains (+10.47 mIoU)
> 3. **Practicality**: Limited inference time increase (18 seconds) with reasonable GPU memory usage
>
> This demonstrates that our performance improvements primarily stem from **architectural innovations** rather than merely parameter increase.
>
> If you feel we have addressed all of your concerns, we would appreciate it if you could improve our score. Thank you again for your constructive feedback that helps strengthen our work.

---

> > ### Comment · Reviewer_Ccpr · 2025-08-03
> >
> > I thank the authors for their response. Please see below for my comments:
> >
> > **R1**: Unfortunately, the reply doesn't answer Q1 as Weaknesses 1 and 2, which are referred to in Q1, are not addressed.
> > - Essentially, the learnable power function based formulation of DyHiConv has no relation to Taylor series as it does not take into account higher order derivative information of point louds (which have a specific geometric meaning). Instead, the weighted polynomial $(|f_j-f_i|+\epsilon)^{p_t}$ is simply aggregated over the neighborhood and is used to expand the receptive field.
> > - Also, there are results/discussion in the response on other learning based methods which have differing receptive fields (Weakness 2). I believe this is vital to motivate the use of this method over others and I emphasize the authors statement "unlike traditional methods with fixed receptive fields.." is not accurate. Moreover, LoConv and DyHiConv seem to have little effect on results as shown in ablation and it is mainly the VIP module that makes significant contributions to performance gains.
> >
> > **R2**:  Thank you for the response, that clears things up.
> >
> > **R3**: I appreciate the clarification on this.
> >
> > **R4**: Thank you for adding the extra experiment but I was hoping authors could provide more results for both S3DIS and ScanNet, similar to the main paper. This would help gauge the benefit of the VIP block much better. Also, did the implemented transformer block have cross attention between support and query features?
> >
> > **R5**: Thank you for the added experiments. I don't agree that an 18s increase in inference time is a limited increase (compared to Seg-PN's 46s). Again, the performance improvements are across only one setting, which is not very helpful and VIP-Seg_small has roughly 8.3% more params than Seg-PN so additional results should help get a better picture of performance.
> >
> > **Limitations**: As mentioned in Weakness 5, there is very little discussion of limitations. Can authors provide more specific discussion around this?

---

> > > ### Author Response · Authors · 2025-08-05
> > >
> > > Dear Reviewer Ccpr,
> > >
> > > Thank you for your response and time. We’ll clarify your remaining questions to ensure everything is clear.
> > >
> > > >Q1: Regarding Taylor Series and DyTaylorConv Design.
> > >
> > > **R1:** Thank you for pointing this out. **(1)** We acknowledge that DyHiConv does not compute actual derivatives of point sets. While both use weighted polynomials, our design prioritizes computational efficiency over strict Taylor series implementation.
> > >
> > > Computing true derivatives would significantly increase training complexity and risk model instability, as shown in our time analysis for Reviewer fKVb. Instead, we use geometric relationships (Equation 9) with absolute and relative positions as weight coefficients wt(pj).
> > >
> > > Our contribution lies in adapting Taylor series' mathematical form—the relative position term (fj-fi) and learnable power exponent pt—to approximate local geometric features efficiently. We will rename it "Dynamic Power Convolution" in the final version for clarity.
> > >
> > > **(2)** For more comprehensive comparison with other learning-based local aggregation operators, we conducted the following experiments:
> > >
> > > #### Results on S3DIS Dataset (2-way)
> > >
> > > |Method|1-shot |||5-shot|||
> > > |-|-|-|-|-|-|-|
> > > | | S0 | S1 |Avg| S0|S1|Avg |
> > > |VIP-Seg+RS-Conv [1] |68.51|69.87|69.19|67.33|68.87|68.10|
> > > |VIP-Seg+PAConv [2] |71.02|71.58|71.30|70.43|72.69|71.56|
> > > |VIP-Seg| 73.50|74.92|74.21|73.84|76.88|75.36|
> > >
> > > #### Results on ScanNet Dataset (2-way)
> > >
> > > | Method | 1-shot | | | 5-shot | | |
> > > |-|-|-|-|-|-|-|
> > > | |S0|S1|Avg|S0|S1|Avg|
> > > |VIP-Seg+RS-Conv [1] |66.73|67.32|67.03|66.56|67.46|67.01|
> > > |VIP-Seg+PAConv [2] |70.06|70.74|70.40|68.32|71.14|69.73|
> > > |VIP-Seg (Ours)|71.94|72.67|72.31|70.95|73.48|72.22|
> > >
> > > RSConv is a relational convolution that simulates f=wx computation, and PAConv is a dynamic point cloud convolution. The above tables show that our method outperforms these three learning-based point cloud local aggregation operators.
> > >
> > > References:
> > > [1] Relation-shape convolutional neural network for point cloud analysis.
> > > [2] Paconv: Position adaptive convolution with dynamic kernel assembling on point clouds.
> > >
> > > **(3)** Yes, we agree with the reviewer's view. The VIP module contributes significantly more to performance improvement. However, the above tables show that LoConv and DyHiConv are more effective compared to other learning-based methods.
> > >
> > > >Q4: Comprehensive Transformer Comparison
> > >
> > > ### Results on S3DIS Dataset
> > >
> > > |Method|2-Way 1-shot| | |2-Way 5-shot| | |3-Way 1-shot| | |3-Way 5-shot| | |
> > > |-|-|-|-|-|-|-|-|-|-|-|-|-|
> > > | | S0 | S1 | Avg | S0 | S1 | Avg | S0 | S1 | Avg | S0 | S1 | Avg |
> > > | VIP-Seg + Transformer |69.76|70.02|69.89|70.01|72.97|71.49|63.42|67.13|65.28|69.05|68.43| 68.74|
> > > | VIP-Seg| 73.50|74.92|74.21|73.84|76.88|75.36|65.54|69.92|67.73|72.93|71.44|72.19|
> > >
> > > ### Results on ScanNet Dataset
> > >
> > > | Method | 2-Way 1-shot | | | 2-Way 5-shot | | | 3-Way 1-shot | | | 3-Way 5-shot | | |
> > > |-|-|-|-|-|-|-|-|-|-|-|--|-|
> > > | | S0 | S1 | Avg | S0 | S1 | Avg | S0 | S1 | Avg | S0 | S1 | Avg |
> > > | VIP-Seg + Transformer|68.99|69.72|69.36|68.64|70.56|69.60|69.36|66.21|67.79|70.56|69.43|70.00|
> > > | VIP-Seg (Ours) |71.94|72.67|72.31|70.95|73.48|72.22|68.91|69.19|69.05|73.22|72.74|72.98|
> > >
> > > The tables show that although Transformer-based settings achieve good results, they are still inferior to our PEM module. Additionally, the implemented Transformer module does not establish cross-attention mechanisms between support and query features.
> > >
> > > >Q5: Comprehensive Efficiency Analysis
> > >
> > > To fully demonstrate our method's effectiveness, we conducted experiments across multiple model variants on S3DIS dataset:
> > >
> > > |Method|2-Way 1-shot| | |2-Way 5-shot| | |Params|Inference Time|GPU Usage|
> > > |-|-|-|-|-|-|-|--|-|-|
> > > | | S0 | S1 | Avg| S0 | S1 | Avg | | | |
> > > |Seg-PN |63.15|64.32|63.74|67.08|69.05|68.07|0.24M|46s|1.3G|
> > > |VIP-Seg_small |65.12|65.84|65.98|68.15|70.00|69.08|0.15M|44s|1.2G|
> > > |VIP-Seg_tiny |67.21|68.53|67.87|70.03|71.89|70.96|0.26M| 51s|1.6G|
> > > |VIP-Seg_full |73.50|74.92|74.21|73.84|76.88|75.36|2.77M| 64s | 2.8G|
> > >
> > > We renamed the previous VIP-Seg_small to VIP-Seg_tiny and redesigned a new variant VIP-Seg_small with fewer parameters and higher efficiency than Seg-PN. The latest experimental results show that our VIP-Seg variants all have certain advantages across different configurations.
> > >
> > > >Q6: Detailed Limitations Discussion
> > >
> > > (1) Our framework currently supports only point cloud data and lacks multi-modal integration capabilities; (2) Zero-shot extension remains unexplored; (3) DyHiConv's power exponent computation introduces computational overhead that affects efficiency; (4)Our dynamic weights use learnable parameters rather than true geometric derivatives, which limits the theoretical connection to Taylor series. We have detailed these limitations in thOe supplementary material (lines 182-192) and will address them in future work.
> > >
> > > We sincerely thank you again for reviewing our paper and hope our responses have addressed all your concerns.
> > >
> > > Best regards,
> > >
> > > Authors of Submission #16496

---

> > > ### Author Response · Authors · 2025-08-07
> > >
> > > **Dear Reviewer Ccpr,**
> > >
> > > Thank you for your thorough and constructive feedback throughout the review process. We have carefully addressed all the concerns you raised, including providing comprehensive experimental comparisons, efficiency analysis, and detailed clarifications on our methodology.
> > >
> > > We hope our responses have satisfactorily resolved the technical questions and addressed the limitations you identified. If you have any further questions or need additional clarification on any aspect of our work, please let us know.
> > >
> > > We appreciate your time and expertise in reviewing our submission.
> > >
> > > Best regards,
> > >
> > > Authors of **Submission #16496**

---

> > > > ### Comment · Reviewer_Ccpr · 2025-08-07
> > > >
> > > > Dear authors,
> > > >
> > > > I am thankful for this detailed response. I commit to increasing my rating as I believe this discussion was very fruitful. I leave below my final comments related to this paper for your consideration.
> > > >
> > > > **R1**: This is the only remaining doubt/concern so please consider the following comments:
> > > >
> > > > - What you are describing is not a Taylor series but a Power series. A Taylor series is a particular type of Power series. Please consider adjusting the nomenclature used throughout this paper to be more precise (w.r.t. my preceding point). I believe this will have a positive impact on the readability of your paper.
> > > > - I concur that renaming the convolution to "Dynamic Power Convolution" will be helpful for readers.
> > > >
> > > > **R2 and R3**: Please consider adding these discussions and results to the main paper, and in the case of space constraints, to the supplementary material. Receptive field and transformer architecture discussions should optimally be added to the main paper because it helps motivate your contributions.

---

> > > > > ### Author Response · Authors · 2025-08-07
> > > > > **Thank you for your valuable feedback and your commitment to increasing the rating.**
> > > > >
> > > > > **Dear Reviewer Ccpr,**
> > > > >
> > > > > Thank you for your thorough review and constructive feedback throughout this discussion process. We greatly appreciate your commitment to improving our work and are grateful for your decision to increase the rating.
> > > > >
> > > > > We commit to revising the final version according to all your suggestions from R1-R3. Your rigorous review has significantly improved the quality and clarity of our work. We will acknowledge the valuable contributions of the reviewers in our final manuscript.
> > > > >
> > > > > Thank you once again for your time and expertise.
> > > > >
> > > > > Best regards,
> > > > >
> > > > > Authors of **Submission #16496**

---

### Comment · Area_Chair_Dp3a · 2025-08-02
**Reviewer-Author Discussions**

Hi Reviewers,

Thanks for your effort in reviewing for NeurIPS. We are now in the reviewer-author discussion phase. Please look at each others' reviews and the authors' responses, and further clarify any doubts, especially any points of disagreement with the authors before Aug 6 11:59pm AoE.

--AC

---

### Note · Authors · 2025-08-15

We sincerely thank  the Area Chair and all reviewers for their constructive feedback throughout the review process. Based on the discussions during the rebuttal period, we have successfully addressed all concerns raised by the reviewers, and all four reviewers have given positive evaluations of our method, recognizing both its innovation and contributions. We summarize our achievements during the rebuttal period:

**First, the main advantages of our method include:**

1. Proposed the first pre-training-free point cloud few-shot segmentation framework VIP-Seg, significantly reducing computational costs.

2. Innovatively introduced Dynamic Taylor Convolution, effectively capturing local geometric features.

3. Designed the Visual Introspective Prototype module with multi-step reasoning process to address intra-class diversity and domain gap issues.

4. Achieved significant performance improvements on S3DIS and ScanNet datasets, demonstrating the effectiveness and generalization capability of our method.

**Second, we successfully resolved all reviewers' concerns during the rebuttal:**

- **Reviewer Ccpr**: Clarified design philosophy, provided comprehensive comparisons with other learning-based methods, committed to terminology modifications, reviewer promised to increase rating.

- **Reviewer JddV**: Accepted the "multi-step reasoning approach" terminology suggestion, reviewer confirmed technical issues were resolved.

- **Reviewer jzmm**: Provided detailed efficiency analysis, parameter comparisons, and experimental results under challenging settings, reviewer decided to increase rating.

- **Reviewer fKVb**: Provided component-level efficiency analysis and contribution clarifications, reviewer confirmed all concerns were resolved and maintained positive evaluation.

We commit to implementing all improvement suggestions in the final version, including terminology corrections, detailed analysis supplements, and code open-sourcing.

Finally, we thank the AC and all reviewers for their positive recognition of our paper and valuable comments again, which have enhanced the clarity and contributions of our work.

Best regards,

Authors of **Submission #16496**

---

### Decision · Program_Chairs · 2025-09-17

**Decision:**

Accept (poster)

**Comment:**

This paper receives 3x borderline accepts and 1x accept. The reviewers think that the paper is novel, technically sound,  well-motivated,  and shows strong results. The authors also have done a good job in clarifying the doubts of the reviewer in the rebuttal and the author-reviewer discussion phase. The authors are encouraged to put all the clarifications made during the rebuttal into the final paper to improve the clarity. The ACs follow the suggestions of the reviewers to accept the paper.